# Interaction of TLR4 and TLR8 in the Innate Immune Response against Mycobacterium Tuberculosis

**DOI:** 10.3390/ijms22041560

**Published:** 2021-02-04

**Authors:** Shruthi Thada, Gabor L. Horvath, Mario M. Müller, Nickel Dittrich, Melanie L. Conrad, Saubashya Sur, Abid Hussain, Karin Pelka, Suman Latha Gaddam, Eicke Latz, Hortense Slevogt, Ralf R. Schumann, Sanne Burkert

**Affiliations:** 1Institute of Microbiology, Infectious Diseases and Immunology, Charité Universitätsmedizin Berlin, Corporate Member of Freie Universität Berlin, Humboldt-Universität zu Berlin and Berlin Institute of Health, 12203 Berlin, Germany; shruthi.thada@gmail.com (S.T.); nickel.dittrich@gmail.com (N.D.); melanie.conrad@charite.de (M.L.C.); ralf.schumann@charite.de (R.R.S.); 2Bhagwan Mahavir Medical Research Centre, Hyderabad 500004, India; sumanlathag@yahoo.com; 3Institute of Innate Immunity, University Hospitals Bonn, University of Bonn, 53127 Bonn, Germany; horvathg@uni-bonn.de (G.L.H.); eicke.latz@uni-bonn.de (E.L.); 4Host Septomics, ZIK Septomics, Jena University Hospital, 07745 Jena, Germany; mario.mueller1@med.uni-jena.de (M.M.M.); hortense.slevogt@med.uni-jena.de (H.S.); 5Postgraduate Department of Botany, Life Science Block, Ramananda College, Bishnupur 722122, India; saubashya@gmail.com; 6Department of Biotechnology and Bioinformatics, University of Hyderabad, Hyderabad 500046, India; mdabid19@gmail.com; 7Broad Institute of MIT and Harvard, Cambridge, MA 02139, USA; kpelka@broadinstitute.org; 8Center for Cancer Research, Massachusetts General Hospital, Boston, MA 02129, USA; 9Department of Genetics, Osmania University, Hyderabad 500007, India; 10Department of Internal Medicine III, Division of Infectious Diseases, University Hospital of Ulm, 89081 Ulm, Germany

**Keywords:** TLR4, TLR8, tuberculosis, SNP analysis, heterodimerisation

## Abstract

The interaction and crosstalk of Toll-like receptors (TLRs) is an established pathway in which the innate immune system recognises and fights pathogens. In a single nucleotide polymorphisms (SNP) analysis of an Indian cohort, we found evidence for both TLR4-399T and TRL8-1A conveying increased susceptibility towards tuberculosis (TB) in an interdependent manner, even though there is no established TLR4 ligand present in *Mycobacterium tuberculosis* (Mtb), which is the causative pathogen of TB. Docking studies revealed that TLR4 and TLR8 can build a heterodimer, allowing interaction with TLR8 ligands. The conformational change of TLR4-399T might impair this interaction. With immunoprecipitation and mass spectrometry, we precipitated TLR4 with TLR8-targeted antibodies, indicating heterodimerisation. Confocal microscopy confirmed a high co-localisation frequency of TLR4 and TLR8 that further increased upon TLR8 stimulation. The heterodimerisation of TLR4 and TLR8 led to an induction of IL12p40, NF-κB, and IRF3. TLR4-399T in interaction with TLR8 induced an increased NF-κB response as compared to TLR4-399C, which was potentially caused by an alteration of subsequent immunological pathways involving type I IFNs. In summary, we present evidence that the heterodimerisation of TLR4 and TLR8 at the endosome is involved in Mtb recognition via TLR8 ligands, such as microbial RNA, which induces a Th1 response. These findings may lead to novel targets for therapeutic interventions and vaccine development regarding TB.

## 1. Introduction

The recognition of potentially pathogenic microorganisms followed by an inflammatory response of the host is regulated by the immediate reaction of the innate immune system [1]. Activation of this evolutionary older system is also crucial for an efficient function of the second arm of the immune response present only in vertebrates, the acquired immune system. Antigen-presenting cells (APCs) are activated and migrate to the lymph nodes, where they bridge the innate and adaptive immune systems by presenting antigens, leading to the generation of an efficient antibody response [2]. Pattern Recognition Receptors (PRRs), which have been identified and structurally characterised over the last 20 years, play a major role in mounting an effective innate immune response by recognising the presence of pathogens via Pathogen-Associated Molecular Patterns (PAMPs) [3]. Toll-like receptors (TLRs) are one important subgroup of PRRs mainly present on APCs such as alveolar macrophages and dendritic cells (DCs) [4]. Several TLRs located in the cell surface membrane have the main function of recognising bacterial cell wall compounds, internalising the microbe, and activating a nuclear factor kappa-light-chain-enhancer of activated B cells (NF-κB)-mediated inflammatory response. Others are expressed within the endosomal membrane and act to recognise microbial nucleic acids, inducing type I interferons (IFNs) [5]. TLRs act as dimers, and while most receptors organise as homodimers, some have been structurally analysed as functional heterodimers [6]. For example, the plasma membrane located TLR2 can form a heterodimer either with TLR1 or -6, resulting in a change in its ligand-binding capacity and a more specific response to Gram-positive bacteria [7].

Tuberculosis (TB) is caused by *Mycobacterium tuberculosis* (Mtb), and both the cell wall composition and the immune response elicited by Mtb within the host are unique. Certain cell wall compounds of Mtb have been described to interact with PRRs located mainly on the outer cell membrane, particularly TLR2/1 [8,9]. Furthermore, TLR4 has been suggested to recognise mycobacterial lipoarabinomannan (LAM) and lipomannan (LM), although it has remained unclear how a ligand so distinct from the typical TLR4-ligand lipopolysaccharide (LPS) can interact with TLR4 [10]. TLR4 can be localised either on the cell surface or the endosomal membrane. For the interaction of TLR4 with LPS, MD-2 and CD14 are required [11,12]; however, for other ligands, this might not be the case. Furthermore, MD-2 and CD14 promote the LPS-induced endocytosis of TLR4 [13].

Interestingly, the endosomal localisation of TLR4 changes the utilisation of adapter molecules, leading to a differentiated inflammatory response. TLR4 activated within the endosome does not recruit its standard adaptor protein myeloid differentiation primary response 88 (MyD88) but instead, it recruits TIR-domain-containing adapter-inducing interferon-β (TRIF), which activates the transcription factor IFN regulatory factor (IRF)-3 and thereby changes the immune response from NF-κB-dependent cytokines to type I IFNs [14,15]. Generally, TLR4 activation promotes a T helper (Th)1 response by activating APCs and promoting DC maturation, inducing interleukin (IL)-12, tumour necrosis factor (TNF)α, IFNγ, major histocompatibility complex class (MHC)-II, CD80, and 86, and NO, as well as enhancing phagocytosis and inhibiting IL-10 production [16,17,18,19,20]. Mtb is known to strongly inhibit the expression of TLR4 and both its adaptor proteins TRIF and MyD88 [21,22]. Furthermore, mycobacteria evade PRR recognition of the cell wall by persisting intracellularly in macrophageal phagosomes. Therefore, once Mtb is internalised, intracellular PRRs must assume immunosurveillance. Recently, evidence has accumulated that mycobacterial nucleoside recognition both in the endosome (e.g., by TLR8) and within the cytoplasm is important for an effective immune response [23,24,25]. TLR8 is an endosomal receptor recognising uridine-rich and short ssRNA mainly expressed in macrophages and myeloid DCs [26,27]. Activation leads to an induction of NF-κB via MyD88, resulting in the production of IL-12, TNFα, and IFNγ, as well as the induction of type I IFNs through IRF5 and IRF7 [28]. Thus, a Th1 response is promoted. Similarly to TLR4, mycobacteria have developed mechanisms to impair the function of endosomal PRRs by inhibiting endosomal acidification [29,30].

Evidence for both TLR4 and TLR8 being involved in TB pathogenesis comes from clinical trials assessing the frequency of single nucleotide polymorphisms (SNPs) in the TLR4 and TLR8 genes in TB patients compared to healthy controls. Increased susceptibility toward TB in individuals carrying TLR4 SNPs has been described for the variants Asp299Gly (rs4986790) and Thr399Ile (rs4986791) [31]. For TLR8, susceptibility has been associated with the less functional variant of TLR8 Met1Val (rs3764880) [23,32]. In this study, we confirm the evidence for TLR4 playing a role in TB immunity and hypothesise that the endosomal cooperation of TLR4 and TLR8 by forming a heterodimer modulates the immune response to Mtb.

## 2. Results

### 2.1. Cohort Characteristics and Genotyping

We analysed a TB cohort from an unmatched case-control study that was previously described for genetic susceptibility (Appendix A, [23,24,33,34]). The cohort consisted of 346 TB patients and 301 controls. TB patients were either diagnosed with pulmonary (224 patients, PTB) or extrapulmonary TB (121 patients, EPTB). Additionally, there were 95 relapse cases. There were significantly more females among the patients (58.9%) than controls (50.9%) and TB patients, on average, were younger (25.5 years) than controls (32.9 years). Among the relapses, 49.5% were female, and the mean age was 30.6 years. More controls than patients had received Bacillus Calmette–Guérin (BCG) vaccine in their childhood (82.9% and 48.8% respectively). Relapse cases had the lowest mean body mass index (BMI, 16.49), followed by pulmonary (16.6) and extrapulmonary cases (19.9), and controls (24.3). Relapse cases also had the lowest portion of BCG positives (36.92%) in comparison to primary TB cases (48.8%) and controls (82.9%).

Regarding SNP distribution, TLR4-Thr399Ile (cytosine (C)>thymine (T)) and TLR4-Asp299Gly (adenine (A)>guanine (G)) were not fully linked (cosegregation in only 73%), unlike among Caucasians (Appendix A). TLR4-399T was more frequent among TB patients than controls, and there was evidence that it conveyed susceptibility towards TB (OR = 1.97 [1.15–3.37]; *p* = 0.013; Table 1, for full model Appendix A). There was no evidence for effect modulation by BCG vaccination (*p* = 0.392). There was also no evidence for a different distribution of TLR4-A299G alleles between TB patients and controls (OR = 0.72 [0.49–1.07], *p* = 0.101), nor for an impact of allele distribution on the site of manifestation (PTB or EPTB) for TLR4-C399T (OR = 0.85 (0.51–1.41), *p* = 0.523) or TLR4-A299G (OR = 1.24 (0.74–2.05), *p* = 0.413). As we previously reported [23], TLR8-1A was associated with a susceptibility towards being a TB case (OR = 1.68 (1.08–3.63); *p* = 0.022; Table 1), with weak evidence for an interaction between BCG and TLR8-Met1Val (A > G) (*p* = 0.071, Appendix A). Interestingly, the susceptibility conveyed by TLR8-1A was only seen in individuals carrying the TLR4-399T allele (OR= 1.97 (1.15–3.37), *p* = 0.013); among homozygote TLR4-399C individuals, no impact of TLR8-1A on TB disease was observed (OR = 1.19 (0.52–2.72); *p* = 0.681, Table 1). Notably, there was no evidence for the statistical interaction of the two SNPs in primary TB (*p* = 0.363) or relapse (*p* = 0.750) cases.

Regarding relapses, we did not observe a significantly different distribution between primary TB patients and relapse cases regarding TLR4-A299G (OR = 0.80 (0.49–1.32), *p* = 0.381) or TLR4-C399T (OR = 1.36 (0.81–2.28), *p* = 0.245, Table 2). However, there was also strong evidence for an increased risk for being a relapse case associated with TLR8-1A (OR = 1.99 (1.03–3.82); *p* = 0.006, Appendix A). Regarding a potential interaction of TLR4 and TLR8, the same pattern as above was observed, namely that TLR8-1A conveyed susceptibility to TB depending on TLR4-399T although with only weak evidence (OR = 2.90 (0.87–9.59), *p* = 0.069; Table 2). Again, there was no evidence for formal statistical interaction (*p* = 0.750). Nevertheless, this observation led us to suspect that there might be an interaction on the molecular level between TLR4 and TLR8, and we further investigated this in different in silico and in vitro systems.

### 2.2. Docking Outcome

To evaluate the possible structural implications of an amino acid residue change in TLR4 at position 399, we performed in silico analysis. Furthermore, we performed molecular docking studies to investigate the idea of interaction between TLR4 and TLR8. The docking outcome revealed that the TLR4-399C molecule (threonine at position 399) could undergo heterodimerisation with TLR8 in presence of the agonistic ligand R848 (Figure 1). Threonine-399 and serine-400 residues of TLR4 could link to the TLR8 molecule by hydrogen bonds of 2.12 and 2.24 angstroms, respectively. The major non-ligand residues from TLR8 involved in the hydrophobic contacts were Tyr353, Gly351, Val378, Ser352, Ile349, and Tyr348. However, the structure of the TLR4-399T molecule (with isoleucine at position 399) did not show this phenomenon. This demonstrated that the ability of TLR4 to form a heterodimer with TLR8 was lost by changing the residue TLR4 residue threonine-399 to isoleucine-399. This change might lead to conformational rearrangements in the protein structure that could alter the ligand-binding capacity and thereby prevent the R848-facilitated formation of a heterodimer with TLR8.

### 2.3. Mass Spectrometry

To further explore the potential interaction of TLR4 and TLR8, we conducted mass spectrometry analysis on co-immunoprecipitation (IP) to find evidence for heterodimer formation. With IP, we pulled down with anti-human influenza hemagglutinin (HA)-TLR8 in human embryonic kidney (HEK)293 cells that were either transfected with TLR4- and with or without TLR8-HA and analysed the samples for the presence of TLR4, TLR8 and other proteins. As expected, we saw a significant difference in TLR8 intensity between samples transfected with or without TLR8-HA. Furthermore, we identified protein unc-93 homolog B1 (UNC93B1) in the proteomic analysis as statistically significantly enriched when co-immunoprecipitating with TLR8-HA in HEK cells (Figure 2), as expected and reported by others [35]. The TLR4 receptor was identified in five out of six TLR8-HA+TLR4 co-transfected cell lysate immunoprecipitations, with two to five unique peptides, but not in any control sample. However, due to fluctuating LFQ intensities for TLR4, co-IP with TLR8-HA, and high variance in the whole dataset of the experiments, the enrichment of TLR4 in TLR8-HA co-transfected cells immunoprecipitated with anti-HA did not reach statistical significance after Benjamini–Hochberg correction (TLR4 log2 fold change = 4.27 and *p* > 0.05 after B.H. adjustment; before *p* = 0.001). The successful identification of a TLR4 peptide in TLR4/8 co-IP samples by higher-energy collisional dissociation spectrum is shown in Appendix A. Altogether, the data indicate that TLR8 can interact with TLR4 in HEK cells overexpressing both receptors.

### 2.4. Co-Immunoprecipitation

For further investigation of the potential interaction of TLR4 and TLR8, we performed co-IP experiments with TLR4 variants. HEK293XL/hTLR8-HA+UNC93B1 cells were transiently transfected with TLR4–mCherry–myc 399C or 399T followed by stimulation with LPS, R848, LPS+R848, and Mtb RNA, or left untreated for 2 h. First, the lysates were immunoprecipitated with anti-HA antibody to pull down TLR8 and then immunoblotted with either anti-TLR4 or anti-HA (for indirect blotting of TLR8) antibody (Figure 3). As expected, without transfection for TLR8, neither TLR4 nor TLR8 were identified in the precipitates. Interestingly, TLR4 was identified in all cells transfected with both TLR4 and TLR8, even in unstimulated cells. The successful blot of TLR4 after precipitation for TLR8 indicated an interaction of the two TLRs. Of note, adding MD2/CD14 to the transfection abolished this effect (Appendix A). In hTLR8HA+UNC93B1 cells co-transfected with TLR4-399C (the variant that we have identified to be able to interact with TLR8), R848 stimulation resulted in higher TLR4 band intensities as compared to other stimulants (Figure 3B). Adding LPS to R848 decreased TLR4-399C band intensity. Comparing the genotypes of TLR4 T399C by the quantification of band intensities, 399T-transfected cells exhibited less TLR4 band intensities upon stimulation with R848 and Mtb RNA, although both input and unstimulated cells showed higher band intensities for 399T than 399C (Figure 4).

As a control, we performed the same experiment with hTLR7FLAG instead of hTLR8HA, and no TLR4 was found after IP for FLAG (Appendix A). In order to check different species, we repeated the experiment with Rhesus and C. atys TLR4 FLAG-tagged along with HEK293XL/hTLR8-HA+UNC93B1 with the result of successful identification of TLR4 in the co-IP, which is similar to hTLR4 (Appendix A). Altogether, data from co-IP supported the data from modelling, indicating that TLR4 and TLR8 interact at the endosomal level, particularly in cells co-transfected with TLR4-399C and stimulated with R848.

### 2.5. Co-Localisation

HEK293/hTLR8-HA+UNC93B1 cells were transiently transfected with TLR4-mCherry-myc 399C, as well as the accessory proteins gp96, PRAT4A, CD14, and MD2, and stimulated with LPS, R848, LPS+R848, or left untreated for 2 h. Furthermore, ssRNA40 was used for stimulation, as it produced less cell stress due to easier transfection (already complexed with transfection agent) and higher stability, resulting in clearer results comparted to MTB-RNA/LyoVec. As expected, TLR4 was identified both at the outer cell membrane and the endosome, whereas TLR8 was only seen at the latter. Microscopy showed that within the endosome, TLR4 and TLR8 co-localised in all cells transfected with hTLR8HA+ UNC93B1+TLR4-mCherry-myc 399C, irrespective of the stimulant (Figure 5A–E). The number of co-localising endosomes increased in cells stimulated with LPS (*p* < 0.003) or R848 (*p* < 0.001) (Figure 6). For the combination of LPS and R848, an additive effect for co-localisation could be observed (*p* < 0.001). Additionally, cells were treated with dynasore to block the trafficking of TLR4 to the endosome, upon which no co-localisation signal was seen (Figure 5F). Of note, no difference in the results reported was observed for transfection without MD2 and CD14 (data not shown).

### 2.6. Functional Studies

We further explored the functional impact of the interaction of TLR4 and TLR8 using experiments with TLR-transfected HEK293-cells, including the different SNPs of interest. First, the different variants of TLR4 were transiently transfected along with MD2 and assessed for LPS responsiveness (Figure 7A). The ‘wild-type’ variant of TLR4-299A-399C showed the highest NF-κB induction as compared to other variants with a significant difference in comparison to 399-T (*p* < 0.01) but not 299-G (*p* < 0.218). Next, we co-transfected TLR4-variants with TLR8-1A (Figure 7B). NF-κB induction upon LPS stimulation was not detected due to a lack of MD2. Upon stimulation with R848, TLR8-1A co-transfected with TLR4-399C showed a significantly reduced NF-κB induction compared to TLR4-399T (*p* < 0.007). Adding TLR4-399C to TLR8-1A did significantly reduce NF-κB responsiveness (*p* < 0.012), while TLR4-399T failed to do so (*p* < 0.148). Adding MD2 and CD14 to the transfection of TLR4-399C and TLR8 increased LPS and decreased R848 responsiveness (Appendix A). Stimulation with LPS+R848 and mycobacterial RNA showed similar patterns (Appendix A). Of note, as a control, we transfected HEK blue cells with TLR7 and the TLR4 variants of interest and did not observe any differences upon adding TLR4 to TLR7.

To further support the interaction of TLR4 and TLR8 at the endosome, we used CLI-095, which specifically inhibits TLR4 signalling [36]. CLI-095-treated human monocyte-derived macrophages (THP cell line) showed a decreased NF-κB-response upon LPS stimulation (*p* < 0.0001) and increased NF-κB response in the presence of TLR8 ligands R848 (*p* < 0.001) and Mtb RNA (*p* < 0.01; Figure 7C). Furthermore, we blocked endosomal signalling pathways in THP cells with siRNA for MyD88, TRIF-related adaptor molecule (TRAM), or directly TLR4 (Figure 7D). Upon stimulation with TLR8-specific ligands, the NF-κB response was diminished when MyD88, TRAM, and TLR4 were silenced, which was not the case for TLR2-specific stimulation with PAM_3_CSK_4_. Blocking TLR8 signalling by either completely blocking the endosome through treatment with bafilomycin or siTLR8 abolished NF-κB induction, while LPS-stimulated cells did not show any difference (Appendix A). Altogether, experiments with HEK cells and THP cells indicated that the interaction of TLR4 with TLR8 diminishes NF-κB responsiveness upon TLR8 stimulation, which could be partly reversed by blocking TLR4 signalling and completely inhibited by TLR8-specific or total endosomal blockage.

Next, we analysed peripheral blood mononuclear cells (PBMCs) from healthy controls with TLR8-1A, which differed in TLR4-C399T (Figure 7E). Regarding TNFα, individuals with TLR4-399T exhibited higher levels of TNF release upon stimulation with LPS (*p* < 0.001), R848 (*p* < 0.014), or the combination of LPS and R848 (*p* < 0.013). This effect was even more pronounced when looking at IL12p40, with a remarkable difference in induction upon stimulation with both LPS and R848 (*p* < 0.002) (Figure 7F).

Finally, in order to assess different signalling pathways altered by an interaction of TLR4 and TLR8, we performed Western blotting of IRF3 from supernatants of HEK293 cells transiently transfected with different combinations of TLR4 and TLR8 variants and stimulated with TLR4- and TLR8-specific ligands (Figure 8). Transfection with TLR4, even unstimulated, led to high IRF3 expression, which was strongly increased with TLR8 co-transfection, implying that TLR4 together with TLR8 strongly activates type I IFNs, potentially even by spontaneous heterodimerisation in HEK cells. Cells double transfected with TLR4-399C and TLR8 showed higher band intensities than mono-transfected cells (Figure 9A) or cells transfected with 399T and TLR8 (Figure 9C). With TLR4-399T and TLR8, the differences were less pronounced and only in unstimulated and LPS-stimulated cells IRF3 band intensity was higher in double transfected cells (Figure 9B). Altogether, this would support our hypothesis that with TLR4-399C, the heterodimerisation is more likely than with TLR4-399T.

## 3. Discussion

In this paper, we argue for an interaction of TLR4 and TLR8 as a heterodimer, which has functional importance for TB immunity. We came to this conclusion on the basis of (1) finding TLR4 in co-immunoprecipitated lysates of transfected HEK-cells for TLR8, particularly after R848 stimulation, (2) confirming this result with mass spectrometry, (3) seeing co-localisation with confocal microscopy, which increased upon stimulation, (4) finding a significantly enhanced susceptibility towards TB among individuals with TLR4-399T and TLR8-1A, the latter depending on the first, (5) finding evidence in modelling that TLR4-399C can form a heterodimer with TLR8 in the presence of a TLR8-ligand R848, while TLR4-399T might not, and finally (6) seeing in co-IP that with TLR4-399C, R848 stimulation induced a higher TLR4 band intensity than with TLR4-399T. Regarding the functional impact of this interaction, we found that (1) TLR4-399C showed higher NF-κB levels after LPS stimulation compared to TLR4-399T in HEK-cells, (2) in combination with TLR4, TLR8 transfected HEK cells secreted less NF-κB with TLR4-399C, but not with TLR4-399T, (3) blockage of TLR4 in monocyte-derived macrophages led to higher levels of TLR8-induced NF-κB, (4) in PBMCs, TRL4-399T led to more TNFα and IL12p40, and (5) IRF3 seems to be enhanced spontaneously upon co-transfection of TLR4 and TLR8, possibly more so with TLR4-399C.

TLR4 has been suggested to be a main receptor involved in TB immunity by recognising mycobacterial antigens upon which a MyD88- and TRIF-dependent Th1 answer is fostered, although no clear explanation for this has been provided [10]. TLR4 involvement has been supported by mechanisms that Mtb has evolved to avoid the host immune system involving TLR4: mycobacterial anti-inflammatory proteins such as phosphatidylinositol mannosides, LM, and LAM that specifically inhibit TLR4-induced pathways or lead to TLR4-triggered immunosuppression [21,37,38]. Furthermore, Mtb is known to block the acidification and maturation of phagosomes, thereby generally inhibiting host immune receptors that require a low pH to function properly.

TLR4 polymorphisms are differently distributed around the world, and they are attributed to evolutionary pressure from infectious diseases and the migration of mankind over time. TLR4-299G without linkage with TLR4-399T can be found among African populations and is reported to be protective against malaria [39]. However, in Europe, this allele is linked with TLR4-399T [40]. We found that among the Indian population, TLR4-399T can occur as a single non-linked mutation, next to the TLR4-299G/-399T haplotype. TLR4-399T has already been associated with increased TB susceptibility [41,42]. Reduced LPS responsiveness is a known functional implication of this SNP, as we saw in HEKs. However, in THPs and PBMCs, we also surprisingly found hyperresponsiveness. These conflicting data have been reported in the literature, and recently, a mouse model with the human SNPs TLR4-299 and TLR4-399 confirmed that both SNPs contribute to cell hyporesponsiveness [43].

TLR8-1A, by being less functional than TLR-1G, is also associated with TB susceptibility [32]. What we report here, and to our knowledge for the first time, is the direct interaction at the endosomal level of TLR4 and TLR8. Our experiments show that synergy through the simultaneous stimulation of both receptors leads to higher levels of IL-12, and others have shown increased IL-12 in monocyte-derived DCs [29] and a higher expression of antigen-presenting, co-stimulatory molecules on matured DCs [30]. Crosstalk between TLRs to modulate the immune response is an established concept; for instance, studies show that the co-activation of both TLR3 and TLR8 is necessary to achieve a strong IL12p70 answer [27]. It is also known that the co-stimulation of TLR8 and -2 induces a shift towards a Th17-immunity [44]. Another example is the endosomal heterodimerisation of TLR4 and -6 in the presence of the co-receptor CD36 in responses to oxidised LDL during atherogenesis, independent of MD-2 and CD14 [45]. This signalling induced both MyD88- and TRIF-dependent genes. Similarly, in our study, we could show that the heterodimerisation of TLR4 and TLR8 activated both NF-κB- and IRF3-linked pathways.

Co-IP showed that the co-transfection of TLR4 and TLR8 lead even in unstimulated cells to the ability to precipitate TLR4 through TLR8, potentially indicating spontaneous heterodimerisation even without stimulation. Mass spectrometry of the lysed precipitates revealed that UNC93B1, a chaperone required for TLR8 endosomal trafficking, was identified alongside TLR8 [35]. For TLR4, UNC93B1 is not required. From our data, we cannot conclude whether UNC93B1 was merely pulled down alongside TLR8 homodimers or promoted heterodimerisation with TLR4, but this might be a focus of further research.

In confocal studies, we saw co-localisation upon co-transfection with an increase of co-localisation frequency even if only one receptor was stimulated. However, in contrast to co-IP, co-localisation frequency even further increased with double simulation, which was possibly due to the different read-outs and the close proximity of the receptors, not being able to distinguish between co-localisation, homo- and heterodimerisation upon stimulation.

The formation of a heterodimer in co-IP studies and confocal microscopy was observed without MD2 and CD14, although it is the established concept that TLR4/MD2/CD14/LPS is necessary for TLR4 internalisation [46]. This might be due to the experimental set-up, as, in HEK-cells, we delivered TLR4 by transfection directly to the endosome. Interestingly, with MD2 and CD14 along the transfection for TLR4 and TLR8, no TLR4 could not be identified after the precipitation of TLR8 in co-IP-studies, and R848-induced levels of NF-κB in co-transfected HEK-cells decreased with the addition of MD2 and CD14, arguing for an inhibitory effect on the heterodimerisation of TLR4 and TLR8 by the accessory proteins, which is potentially due to the promotion of the homodimerisation of TLR4. In line with this, adding LPS to R848 in co-transfected cells decreased the intensity of the TLR4 band, which is possibly due to the formation of a homodimer of TLR4, thereby decreasing the chances of interaction with TLR8. Inhibiting the interaction of TLR4 and TLR8 with CLI-095 in THPs reversely led to an increase of TLR8-induced NF-κB-levels. In contrast to that, with siRNA, a slight decrease of the NF-κB-signal upon stimulation could be observed, which is possibly due to more cell stress due to the necessary double transfection, as well as, potentially, a less complete inhibition by siRNA compared to CLI-095.

Furthermore, in HEK cells, we could see that mere co-transfection of TLR4 and TLR8 led to an expression of IRF3, which activated the type I IFN axis. This might explain how TLR4 could negatively regulate NF-κB induction by TLR8 activation, namely by shifting the balance from the NF-κB towards the type I IFN pathway. This might also explain why the blockage of TLR4 enhanced NF-κB induction by TLR8 activation.

The most pronounced difference between the TLR4 variants in co-IP was that TLR4-399C, the variant identified as more prone towards heterodimer formation, when undergoing co-transfection with TLR8, showed increased TLR4 band intensity after TLR8 stimulation with R848 compared to both LPS and unstimulated cells. In contrast to that, with TLR4-399T, R848-stimulated precipitation of TLR4 was decreased as compared to after LPS-stimulated and in unstimulated cells, further supporting the notion that the interaction of TLR4 and TLR8 is impaired with the nucleotide change from C to T. For Mtb RNA, the same trend was observed, although to a lesser extent. This might be because stimulation with RNA requires another transfection medium, thereby increasing cell stress, potentially resulting in reduced reactivity. Another reason might be that modelling actually identified R848 as the ligand promoting heterodimerisation, resulting in higher band intensities in our experiments.

In functional studies, we could show that the combination of TLR4-399T and TLR8-1A led to increased NF-κB, TNFα, and IL12p40 levels in PBMCs and THPs upon stimulation with TLR8 ligands in comparison to TLR4-399C, although both TLR4-399T and TLR8-1A individually are each the less functional variants of the SNP. Based on modelling data, with TLR4-399C, heterodimerisation is more likely to occur, possibly leading to more activation of IRF3, thus potentially leading to more type I IFNs but less direct activation of the NF-κB axis. With TLR4-399T, this effect is hindered, thereby producing more NF-κB. Keeping this rationale in mind, we propose that both loss-of-function alleles of TLR8 and -4 convey susceptibility towards TB by altering the balance of the NF-κB and type I IFN axes, possibly more pronouncedly reducing the latter, and that this interaction plays a crucial role in a successful host response against Mtb.

Eliminating TB is a set goal by the WHO by 2030 [47]. In order to achieve this goal, novel intervention strategies are needed, which will be based on a complete understanding of the pathophysiology. Furthermore, individual risk stratification will be important to improve prevention strategies. Therefore, the interaction of TLR4 and TLR8, by offering new treatment targets and understanding individual progression risk, might contribute to eliminating TB in the future. This is particularly needed in the face of increasing incidence of multi-drug resistant TB.

Implications for the importance of TLR4 and TLR8 interaction might be found beyond TB. Endosomal TLRs recognising RNA such as TLR8 play an important role in viral diseases. Regarding the current SARS-CoV-2 outbreak, i.e., it has been suggested that SARS-Cov-2 contains more RNA sequences recognisable by TLR7/8 than SARS-CoV-1, and by that potentially causing more frequently a hyperinflammatory syndrome [48]. Similarly, there have been studies claiming an important role of TLR4 in SARS-CoV-2, as in silico studies identified TLR4 as very likely to respond to spike proteins of SARS-CoV-2 [49]. Interestingly, TLR4 is also associated with cardiometabolic comorbidities such as obesity and hypertension, which are known risk factors for severe COVID with hyperinflammation [50]. Furthermore, TLR4-deficient mice were less susceptible to acute respiratory distress syndrome (ARDS) upon inhalation trauma [51].

Taken together, our data suggest that TLR4 and TLR8 form a heterodimer changing the immune response towards a Th1 balance. Mutations leading to a loss of function of this specific pathway seem to convey susceptibility towards TB. Thus, the interaction of TLR4 and TLR8 might open up new targets for vaccines or therapeutic drugs. Finally, genetic risk stratification may lead to better prevention strategies of individuals at increased risk.

## 4. Materials and Methods

### 4.1. Study Subjects

The cohort of TB patients and controls in Hyderabad has been described before [34]. In brief, the cohort consisted of 346 TB patients with either PTB, EPTB or a relapse, and 301 Controls (HC) including healthy household contacts (HHC). Patients, who attended Free Chest TB Clinic with directly observed treatment surveillance (DOTS) at Mahavir Hospital and Research Centre, Hyderabad, were confirmed with the sputum microscopy for acid-fast bacilli, culture, and chest X-ray or histopathology as per the guidelines of the Revised National Tuberculosis Control Program (RNTCP). Patients with diabetes, hypertension, HIV, and other comorbidities were excluded from the study. Informed consent was obtained from all subjects. The study was approved by the institutional ethics committee of Bhagwan Mahavir Medical Research Centre (BMMRC), Hyderabad, and Charité Medical University Berlin. The German cohort consisted of 853 volunteers, as described earlier [52]. All studies followed the ethical principles of the declaration of Helsinki.

### 4.2. SNP Analysis

Genomic DNA was extracted from whole blood of TB patients and healthy volunteers using a DNA Blood mini kit (Qiagen GmbH, Hilden, Germany) or from buccal swabs using a DNA kit (Qiagen) according to the manufacturer’s protocol. Quantity of DNA was confirmed by NanoDrop and DNA was stored at −20 °C. Functionally relevant SNPs were analysed using Light Cycler Assays (Roche) based on the differentiation of fluorescence signals due to nucleic acid differences and the respective melting curves. Primers used are found in Appendix A.

### 4.3. Modelling and Molecular Docking

Homology model of the human TLR4 with threonine at 399 was determined, using the crystal structure of TLR4 (PDB ID: 4G8A) as a template with MODELLER [53]. The structure 4G8A had isoleucine (I) in position 399. Refinement and quality estimation of the model was carried out using Swiss PDB viewer [54] and SAVES server (https://servicesn.mbi.ucla.edu/SAVES/). The structure of TLR8 (PDB ID: 3W3M) with resiquimod (R848) ligand was obtained from the PDB database (www.rcsb.org). Molecular docking was implemented using PatchDock [55] and FireDock [56]. In this process, transformations of docking elements obtained from PatchDock were given as an input to FireDock. Firedock initially performs coarse refinement followed by refinements and energy-based rankings. Next, it implements chain optimisation to reduce steric clashes [56]. The generated model of TLR4 having threonine (position 399) and the structure TLR8 was used as inputs during docking. Similarly, in another study, the structure of TLR4 (I399) was considered to find out if it formed a dimer with TLR8 in the presence of the ligand R848. The interacting residues were visualised with LigPlot v2.

### 4.4. In Vitro Experiments

#### 4.4.1. Stimulants and Reagents

The stimulants LPS, R848, ssRNA40, and PAM_3_CSK_4_ and the antagonists bafilomycin, polymyxin B (PMB), dynasore, and CLI-095 were purchased from Invivogen (Toulouse, France). The concentration of above stimulants and antagonists were standardised as LPS (10 ng/mL), R848 (2 µg/mL), ssRNA40 (5 µg/mL), PAM_3_CSK_4_ (2 µg/mL), bafilomycin (1 µM), PMB (10 µg/mL), Dynasore (50 µM), and CLI-095 (3 µM). Mycobacterial RNA was extracted from gamma-irradiated Mycobacterium tuberculosis H37Rv (BEI Resources, NR-14819) with InnuPrep RNA Mini Kit (Analytik Jena, Germany). Purity was confirmed by Scandrop analysis (Analytik Jena). A 260 nm/280 nm extinction quotient of 1.9-2.0 was considered pure. For transfection, if not otherwise specified, LyoVec (Invivogen) was used in a 3 µg/100 µL dilution according to protocol.

#### 4.4.2. Mutagenesis

hTLR8-pUno3 and hTLR4-pUno3 plasmid were purchased from Invivogen and hTLR4 mcherry-myc. All these plasmids were mutated with QuikChange II XL Site-Directed Mutagenesis Kit (Agilent Genes, Frankfurt am Main, Germany) according to user’s manual using the primers designed with primerX software (Appendix A). Maxi Prep of mutated and original plasmid was performed with Plasmid Maxi Kit (Qiagen, Hilden, Germany). Successful mutation was confirmed by Value Read sequencing (Eurofins, Ebersberg, Germany), using the primer 5′CTGTAGTCGACGATTGCTGC3′ for TLR8 and 5′AGGTAAATGAGGTTTCTGAGTGA3′ for TLR4 designed with Primer3 software.

#### 4.4.3. Cell Line Experiments

THP NF-κB (Invivogen) cells were harvested in RPMI 1640 + 10% FCS + 100 µg/mL blasticidin. Cells were counted and plated on 96-well plates with 1 × 10^5^ cells/well in 150 µL of RPMI 1640 + 10% FCS Medium. The cells were differentiated to macrophages by using PMA (50 ng/mL) 3 h prior to transient transfection. Cells were treated with or without bafilomycin, PMB, dynasore, CL-095 1 h prior to stimulation of R848, LPS, or Mtb RNA/Lyovec for 18–24 h. Then, 20 µL of supernatant were transferred to QUANTI-Blue detection medium (Invivogen) for SEAP estimation at 620 nm absorbance, corresponding with NF-κB-activity.

Hek Blue Null 1 (Invivogen) cells were harvested in Dulbecco’s Modified Eagle’s Medium (DMEM) + 10% fetal calf serum (FCS) + 100 µg/mL zeocin. 1.5 × 10^5^ cells were distributed in T25-flasks and cultured overnight before transient transfection. After 48 h, cells were counted and plated on 96-well plates with 5 × 10^4^ cells/well in 160 µL of Hek Blue Detection Medium (Invivogen). For stimulation, LPS, R848, a combination of LPS/R848 or mycobacterial RNA/LyoVec was added, making up to 200 µL well volume. After 16 h (unless specified) of stimulation, SEAP levels were measured at 620 nm absorbance each value was normalised to the respective negative control (PBS). HEK 293XL hTLR8 HA cells overexpressing UNC93B1-mCitrine cells were cultured in 24-well plates prior to transient transfection [35].

Note: We have confirmed that there was no LPS contamination in the TLR8 ligands by treating THP monocyte-derived macrophages with PMB specifically blocking LPS stimulation by binding to lipid A of LPS (Appendix A).

#### 4.4.4. Transient Transfection

For Co-IP and confocal microscopy, HEK 293XL hTLR8 HA cells were used. HEK 293XL hTLR8 HA transiently transfected using Extreme Gene 9 (Roche) in a 1:3 ratio according to protocol for 24 h. The following plasmids were used for transient transfection: hTLR4 mCherry-myc and its variant forms, with and without the accessory proteins gp96, PRAT4A, CD14, and MD2 as indicated, and empty plasmid (pUno3). All the plasmid combinations were attained to a final concentration of 3 µg. Regarding the additional set of experiments with monkey TLR4 plasmids, pEF1a_rhesus TLR4 N-FLAG IRES DsRed Express2 and pEF1a_sooty mangabey TLR4 N-FLAG IRES DsRed Express2 (provided by Prof. Dr. Sauter, Ulm) were performed as described above. For functional studies with HEK Blue Null 1 cells, transient transfection as described above was performed with and without MD2 and CD14 as indicated.

THP NF-κB: The cells were transfected using the Amaxa Nucleofector (Amaxa, Cologne, Germany) according to the manufacturer’s protocol (Cell Line Nucleofector Kit V, Program T-08) with 2 μg DNA/10^6^ cells, psiRNA TLR8, psiRNA TLR4, psiRNA MyD88 or psiRNA Ticam2 (plasmid-based siRNA designed by Invivogen).

#### 4.4.5. Immunofluorescence Staining of TLR8HA/Confocal Microscopy

First, 1 × 10^5^ HEK 293XL hTLR8-HA UNC93B1-mCitrine cells/well were seeded in imaging dishes (Ibidi), which was followed by transfection including accessory proteins gp96, PRAT4A, CD14, and MD2 for 48 h and stimulation with various ligands for 2 h. TLR8-HA was stained with anti-HA antibody (Sigma Aldrich, Munich, Germany) at a dilution of 1:200 in PBS containing 1% (*w/v*) bovine serum albumin (BSA) for 1 h at room temperature. Specimens were washed three times with PBS and incubated with anti-rabbit Alexa647 antibody diluted 1:2000 in PBS containing 1% (*w/v*) BSA for 30 min at room temperature. TLR4 plasmid has mCherry fluroprobe. Then, cells were imaged on a Leica SP5 AOBS with SMD confocal microscope, with a 63×, NA 1.20 water-immersion objective, at a lateral resolution of 120 nm. Cell Profiler and Fiji software were used to analyse the co-alocalisation, which was defined as a spatial overlap of fluorescent TLR4- and TLR8-labels indicated by yellow dots [57].

#### 4.4.6. Co-Immunoprecipitation

Per condition, 5 million cells were lysed in 250 µL NP40 lysis buffer for 30–60 min on ice. Lysates were collected by centrifuging at 4000× *g* for 5 min at 4 °C. Then, 30 µL of lysate was saved as a control, and the rest of the lysate was used for IP. IP was performed with HA agarose beads/anti-FLAG-M2 affinity gel (Sigma Aldrich, Munich, Germany) according to the manual. Per IP, 50 µL of the 1:1 suspension of the anti-HA agarose was used, and IP was performed for 2 h at 4 °C shaking. After the last wash, 30 µL 2× Lämmli was added. The lysate collected before IP (Input) served as a positive control, for negative control (NC) lysis buffer without antibody was used.

#### 4.4.7. Western Blot Procedure

Cell lysates were separated by SDS-PAGE and blotted. Membranes were first exposed to Abs specific for anti-TLR4, anti-HA, anti-FLAG, anti-IRF3, and anti-GAPDH (Santa Cruz Biotechnology, Heidelberg, Germany) and subsequently incubated with secondary Abs. Proteins were detected using electrochemiluminescence (ECL) (32106 Pierce™ ECL Western Blotting Substrate). The band intensities were quantified using image J.

### 4.5. Mass Spectrometry (MS)

#### 4.5.1. Sample Preparation

Eluted proteins from IPs (1% SDS in PBS) were reduced with 50 mM of Dithiothreitol (5 min, 95 °C) and diluted with 8 M Urea in 100 mM Tris/HCl pH = 8.0. Buffer exchange and protein digestion was done according to the filter-aided sample preparation protocol [58]. In brief, the reduced proteins were transferred to a 30 kDa Microcon filter unit (YM-30 filter units, Millipore) and centrifuged at 14.000× *g* for 20 min in all consecutive steps, and the flow-through discarded. For washing, 200 µL urea buffer (8 M Urea, 100 mM Tris HCL, pH 8.0) was added, and the centrifugation was repeated. Then, 100 µL of alkylation solution (0.1 M iodoacetamide in urea buffer) was added, and samples were incubated for 20 min in the dark. The alkylation solution was removed by centrifugation followed by two additional centrifugation steps with 200 µL 8 M urea buffer. Afterwards, samples were washed and centrifuged twice with 200 µL 50 mM ammonium bicarbonate buffer. Proteins were digested by the addition of 0.5 µg trypsin in 50 µL digestion buffer (50 mM ammonium bicarbonate). Proteolytic cleavage was allowed for 16 h at 37 °C, and peptides eluted by centrifugation. To collect residual peptides, the centrifugation was repeated twice after the addition of 50 µL ammonium bicarbonate buffer (50 mM). Eluted peptides were dried in a SpeedVac (Thermo Fisher) and reconstituted by adding 20 µL of 0.3% formic acid in water.

#### 4.5.2. Mass Spectrometric and Statistical Analysis

Tryptic peptides were analysed with a Dionex UHPLC (Thermo Scientific) coupled to an Orbitrap Fusion LC-MS/MS system (Thermo Scientific). Full mass spectrometry scans were acquired in the Orbitrap (m/z range 370–1570, quadrupole isolation) at a resolution of 120,000 (full width at half maximum) during a 60 min, non-linear gradient from 2 to 90% acetonitrile/0.1% formic acid. Peptides were fragmented by higher-energy collisional dissociation (HCD, 30% collision energy) and maximum 10 fragment ion spectra were acquired per cycle in the Orbitrap analyser at a resolution of 15,000 using quadrupole isolation (m/z window 1.6). The following conditions were used: spray voltage of 2.1 kV, heated capillary temperature of 275 °C, S-lens RF level of 60%, a maximum automatic gain control (AGC) value of 4 × 10^5^ counts for MS1 with a maximum ion injection time of 50 ms and a maximum AGC value of 5 × 10^4^ for MS2, with a maximum ion accumulation time of 45 ms. A dynamic mass exclusion time window of 5 s was set with a 10 ppm maximum mass window.

All raw files were searched against the human UniProt database (version 05.2016, reviewed sequences) with MaxQuant version 1.5.5.1 (Max Planck Institute of Biochemistry, Germany) [59]. The default parameters were used or set as follows: first search peptide tolerance: 20 ppm, main search peptide tolerance: 4.5 ppm (for MaxQuant); enzyme: trypsin, max. 2 missed cleavages; static modification: carbamidomethylation of cysteine residues; variable modifications: methionine oxidation; min. peptide length: 6, max. peptide mass: 7600 Da. Normalisation was omitted and Label-Free Quantification (LFQ) min. ratio count was set to 1 (unique and razor peptides). Peptide specific match (PSM) and protein false discovery rate was set to 0.01. Label-Free Quantification (LFQ) values of all samples were loaded into Perseus (version 1.5.5.0) [60]. Groups were created, with 6 samples per group: (a) TLR4 with and without stimulation, (b) TLR8+TLR4 with and without stimulation, (c) TLR8 with and without stimulation. The resulting matrix was reduced as proteins were identified as “possible contamination” or “only identified per site”, while ”reverse identified proteins” and “identified in less than 2 samples per group” were discarded. LFQ values were log2-transformed, and missing values imputed by default parameters. The negative logarithmic difference of the means of protein intensities of MS was plotted against the *p*-values from respective *t*-tests.

### 4.6. PBMC Experiments

PBMCs from individuals of the Indian cohort differing in their genotype of TLR8-M1V and TLR4-T399I were isolated with lymphocyte separation medium (LSM 1077 GE Healthcare) according to the user’s manual, seeded at 3 × 10^5^ cells per well on 96-well plates in RPMI 1640 + 10% FCS and left overnight at 370 in a humidified incubator with 5% CO2. Then, PBMCs were stimulated with LPS, R848, a combination of LPS and R848 or mycobacterial RNA/LyoVec. Cytokines were analysed in the supernatants by enzyme-linked immunoassays according to the respective standard manufacturer’s recommendations. TNFα levels were determined after 4 h (BD Pharmingen: 551220, 554511), and IL-12p40 (BD Biosciences) levels were measured after 24 h post-stimulation.

### 4.7. Statistical Analysis

Statistical analysis was performed using STATA 16. Descriptive characteristics were obtained, and for the assessment of differences of basic characteristics, *t*-tests for continuous variables and chi-square for categorical data tests were used. For assessing the odds associated with a specific allele, genotypes were dichotomised, summarising minor alleles, and a logistic regression model was used; 95% confidence intervals are given in square brackets. The main outcome was being a TB patient or a relapse, depending on the context. Gender and age were included a priori, as was BCG status whenever a TLR8-SNP was assessed, based on previous reports. Other variables were included according to the evidence for confounding based on comparison of crude and adjusted OR, using Wald’s test. Final significance testing and tests for effect modification were based on likelihood ratio tests (LRTs). For analysis of functional, Prism (Version 5.01) was used, performing either Mann–Whitney U or T-tests, as appropriate. For analysis of mass spectrometry, Persus software was used.

## Figures and Tables

**Figure 1 ijms-22-01560-f001:**
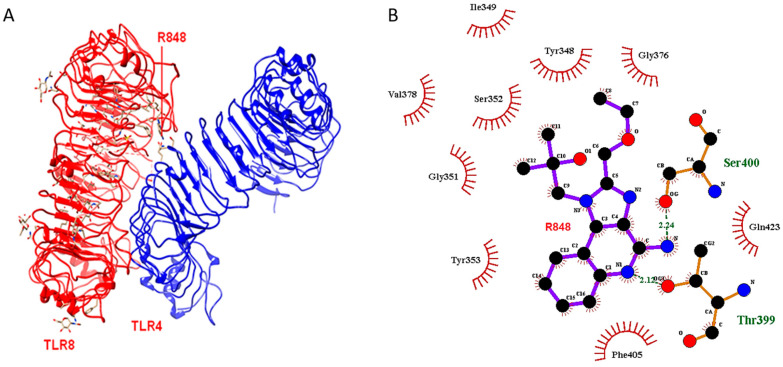
Molecular Docking: (**A**) Toll-like receptor (TLR)4-TLR8 heterodimer mediated by agonistic ligand R848 of TLR8. Both are wild types. (**B**) Ligand plot showing TLR4 variant (Threonin-399, i.e., 399C) interacting with agonistic ligand R848 and assisting in heterodimerisation.

**Figure 2 ijms-22-01560-f002:**
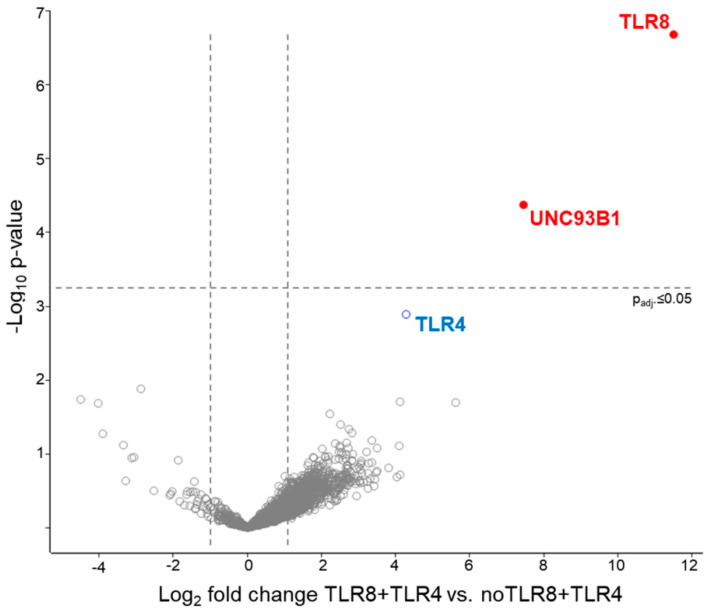
Interaction of TLR8 with TLR4 in HEK cells. Volcano plot of the label-free quantitative MS data plotting the logarithmic difference in protein levels in the HA-immunoprecipitated fraction of TLR8-HA and TLR4 expressing HEK cells and cells expressing TLR4 alone versus the negative logarithmic *p* values of the *t*-test performed of six experiments per group. The dotted lines indicate significance thresholds (fold change ≥ 2 and *p*-value _(Benjamini-Hochberg adj.)_ ≤ 0.05). In red (filled circles), statistically significant differentially abundant proteins, in blue (open circle) TLR4, in grey (open circles), proteins with no statistically significant abundance.

**Figure 3 ijms-22-01560-f003:**
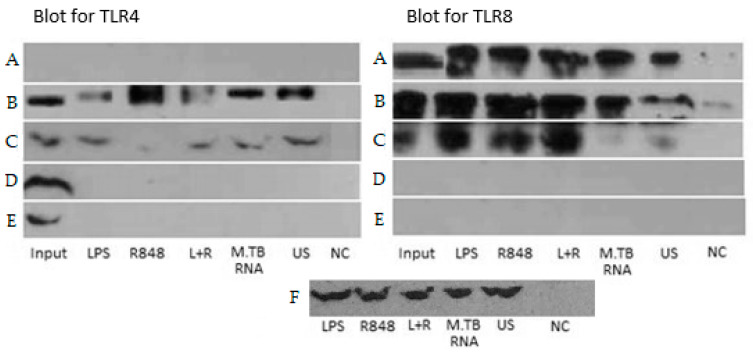
Co-immunoprecipitation. HEK 293 cells transfected as indicated per line, followed by stimulation as indicated per column (2 h for lipopolysaccharide (LPS) (10 ng/mL), R848 (2 µg/mL), LPS and R848 (L+R), 16 h for MTB RNA (1 µg/mL), unstimulated (US) and negative control (NC)). After 2 h, immunoprecipitation procedure was started. The left panel shows immunoprecipitation and -blot with anti-HA antibody (≈110 kDa), indirectly precipitating for TLR8. The right panel shows immunoprecipitation with anti-HA-antibody, followed by immunoblot with anti-TLR4 antibody (100 kDa). (**A**) hTLR8HA+UNC93B1, (**B**) hTLR8HA+UNC93B1+TLR4 399C-mCherry-myc, (**C**) hTLR8HA+UNC93B1+TLR4 399T-mCherry-myc, (**D**) TLR4 399C-mCherry-myc, (**E**) TLR4 399T-mCherry-myc, (**F**) hTLR8HA+UNC93B1 native cells blot with loading control—anti-GAPDH antibody (≈37 kDa). TLR8 was pulled down by anti-HA antibodies and identified in the immunoblot. When HEK cells were co-transfected with both TLR4 and TLR8, TLR4 could be identified in lysates precipitated for HA/TLR8, indicating heterodimerisation.

**Figure 4 ijms-22-01560-f004:**
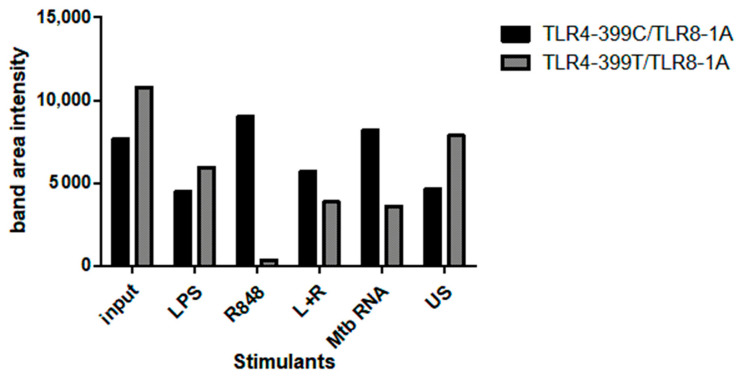
Co-immunoprecipitation quantification. HEK 293 cells transfected with hTLR8HA+UNC93B1+TLR4 399C-mCherry-myc compared with hTLR8HA+UNC93B1+TLR4 399T-mCherry-myc were stimulated with LPS, R848, LPS and R848 (L+R), *Mycobacterium tuberculosis* (Mtb) RNA and unstimulated (US), followed by immunoprecipitation with anti-HA-antibody and immunoblot with anti-TLR4 antibody: In cells stimulated with R8484 and Mtb RNA, the band intensity is higher in hTLR8HA+UNC93B1+TLR4 399C-mCherry-myc transfected cells compared with hTLR8HA+UNC93B1+TLR4 399T-mCherry-myc.

**Figure 5 ijms-22-01560-f005:**
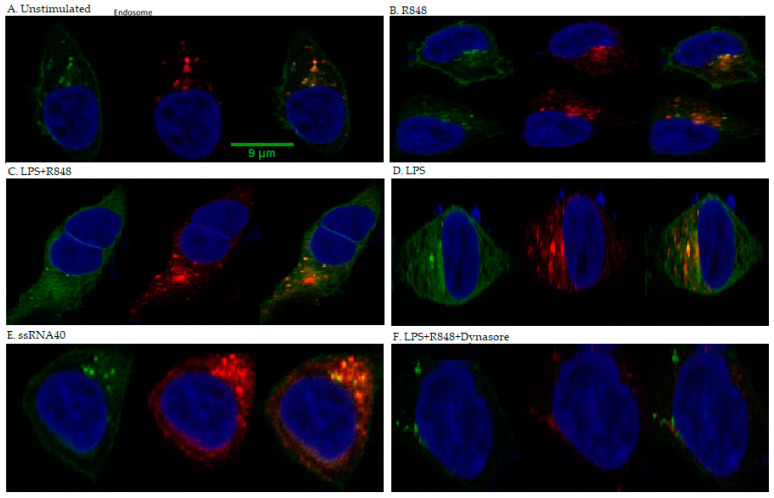
Confocal microscopy. Fluorescence microscopy of HEK293 cells stably transfected with TLR8-HA and transiently transfected with fluorescently tagged TLR4-mcherry along with accessory proteins gp96, PRAT4A, CD14, and MD2. Cells were (**A**) unstimulated or treated with (**B**) R848, (**C**) LPS+R848, (**D**) LPS, (**E**) ssRNA40, and (**F**) LPS+R848+Dynasore for 2 h and stained with an anti-HA Alexa 647-conjugated antibody for TLR8 and Dapi for nuclei. In the false-coloured merged image, double co-localisation of TLR4 (green) and TLR8 (red) in endosomes appears as areas of yellow (arrowhead). Scale bar 9 μm. (**F**) Inhibition of dynamin-dependent endocytosis blocked TLR4-TLR8-triggered downstream pathways by Dynasore.

**Figure 6 ijms-22-01560-f006:**
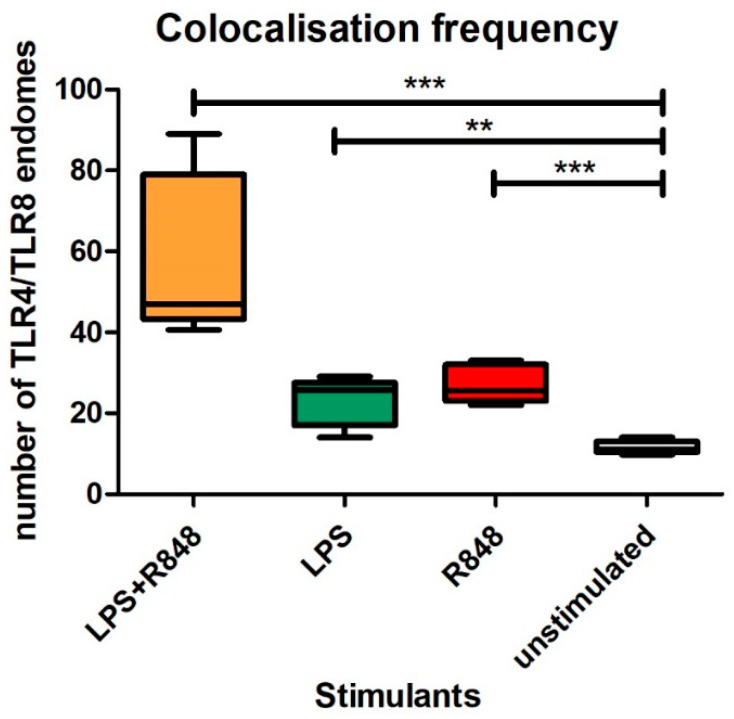
Co-localisation frequency. HEK293 cells stably transfected with TLR8-HA and transiently transfected with fluorescently tagged TLR4-mcherry along with accessory proteins gp96, PRAT4A, CD14 and MD2. Cells were unstimulated or treated with R848 (2 µg/mL), LPS (10 ng/mL), and LPS+R848 for 2 h. Co-localisation was observed under Leica SP5 SMD confocal microscope. The co-localisation frequency increased in cells stimulated with LPS+R848 as compared to LPS, R848, and unstimulated (*p* < 0.009, *p* < 0.01, and *p* < 0.006 respectively). ** *p* ≤ 0.01, *** *p* ≤ 0.001.

**Figure 7 ijms-22-01560-f007:**
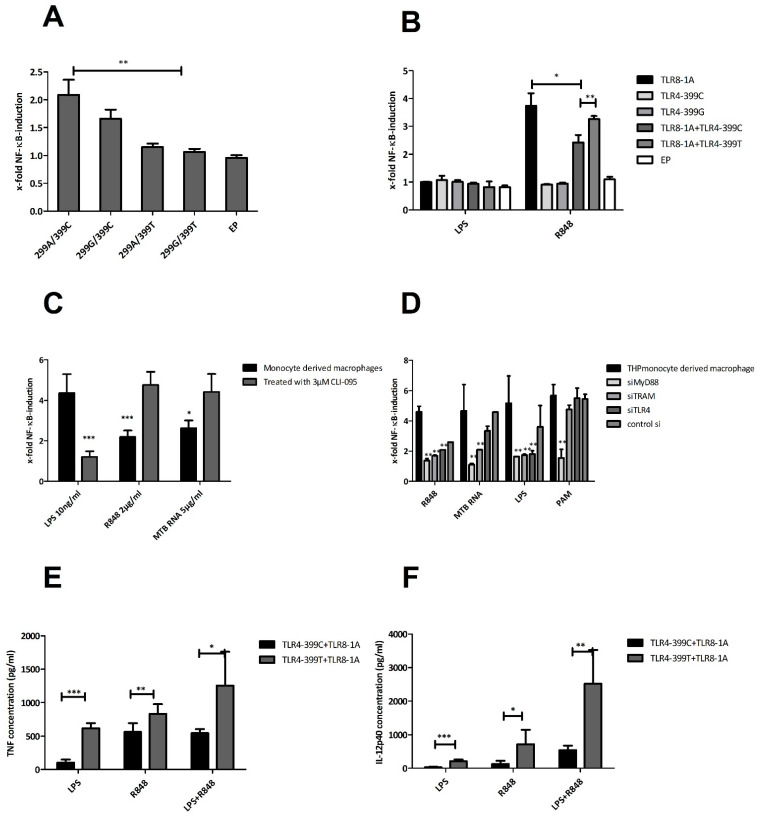
Functional studies within different cell lines to analyse the impact of the SNP TLR4-C399T on the interaction with TLR8. Used cells: (**A**,**B**): HEK293 blue null 1 cells, transiently transfected with TLRs or empty plasmid (EP) as indicated. (**C**,**D**): THP monocyte-derived macrophages. (**E**,**F**): Peripheral blood mononuclear cells (PBMCs) isolated from healthy homozygous volunteers that differed in their status of TLR4-399. Stimulation took place with LPS (100 ng/mL if not otherwise specified), R848 (2 µg/mL if not otherwise specified), Mtb-RNA (5 µg/mL) complexed with Lyovec or PAM_3_CSK_4_ as indicated for 16 h. NF-κB activation was measured by secreted embryonic alkaline phosphatase (SEAP) reporter gene assay, TNFα and IL12-p40 were measured by enzyme-linked immunosorbent assay (ELISA). (**A**) LPS responsiveness of different TLR4 SNPs: Transfection with variants human TLR4 along with human MD2. NF-κB fold induction was significantly raised in cells transfected with TLR4-299A-399C/MD2 compared to TLR4-299G-399C/MD2 (*p* < 0.01) and TLR4-299G-399T/MD2 (*p* < 0.007). (**B**) Co-transfection of TLR8 with TLR4-variants. Without accessory proteins, LPS stimulation was insignificant. TLR8 was stimulated successfully with R848. When adding TLR4-399C, NF-κB levels were significantly lower (*p* < 0.012). The difference between TLR8-1A+TLR3-399C and TLR8-1A-399T was also significant (*p* < 0.007). TLR8-1A and TLR8-1A+TLR4-399T did not show a significantly different response (*p* < 0.147). (**C**) Inhibition of TLR4 signalling with CLI-095. THP monocyte-derived macrophages were stimulated with or without 3 µM CLI-095 (LPS at 10 ng/mL). NF-κB response in the presence of TLR8 ligand R848 (*p* < 0.001), Mtb RNA (*p* < 0.01), and LPS (*p* < 0.001) decreased (**D**) TLR signalling adaptor protein inhibition with siRNA. THP monocyte-derived macrophages were with or without silencing MyD88, TRAM, or TLR4 (LPS at 10 ng/mL, R848 at 5 mg/mL). TLR8 ligand stimulation significantly decreased in presence of siMyD88 and siTRAM (*p* < 0.05). (**E**) TNFα and (**F**) IL-12p40-levels of PBMCs. Individuals with TLR4-399T showed more tumour necrosis factor (TNF)α upon stimulation with LPS (*p* < 0.001), R848 (*p* < 0.014), and LPS+R848 (*p* < 0.032) in comparison to individuals with TLR4-399C. Regarding IL-12p40, there were significantly higher concentrations in individuals with TLR4-399T with LPS (*p* < 0.001), R848 (*p* = 0.016), and LPS+R848 (*p* < 0.002) than 399C. * *p* ≤ 0.05, ** *p* ≤ 0.01, *** *p* ≤ 0.001.

**Figure 8 ijms-22-01560-f008:**
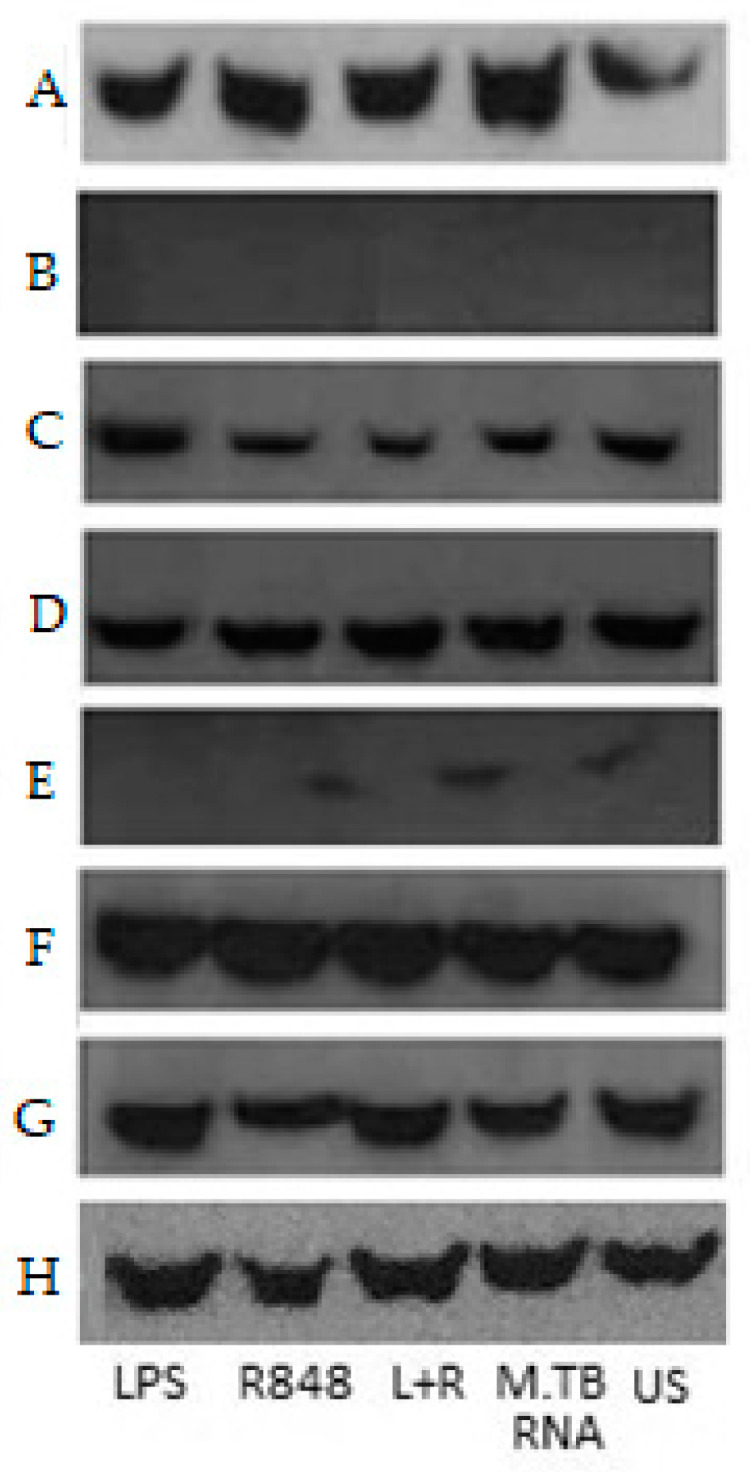
IRF3 Western blotting. (**A**) native HEK293 cells (**B**) HEK293 cells transfected with TLR4-399C/MD2 (**C**) HEK293 cells transfected with TLR4-399C (**D**) HEK293 cells transfected with TLR4-399T (**E**) HEK293 cells transfected with TLR8-1A (**F**) HEK293 cells transfected with TLR4-399C+TLR8-1A (**G**) HEK293 cells transfected with TLR4-399T +TLR8-1A were stimulated with LPS, R8484, LPS+R848 for 2 h and Mtb RNA (16h) or left untreated (US). (**H**) Native HEK293 cells blot with loading control–anti GAPDH antibody (≈37 kDa). IRF3 (≈55 kDa) could be identified in all double-transfected cell lysates.

**Figure 9 ijms-22-01560-f009:**
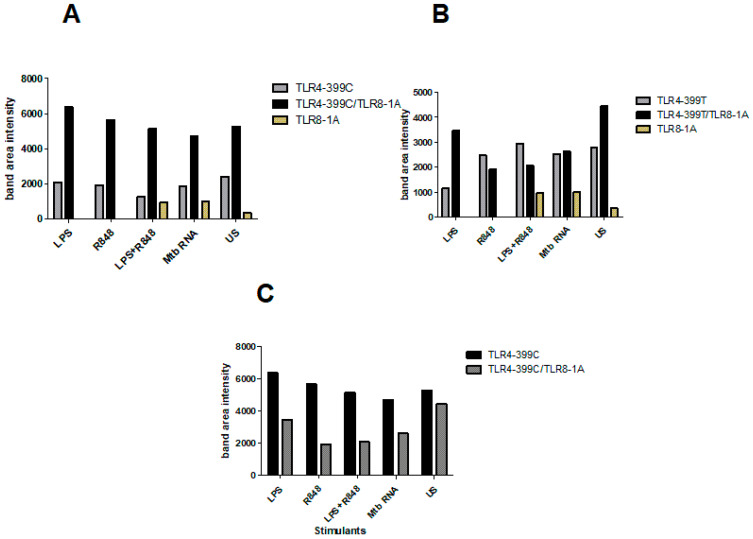
IRF3 Western blot quantification in HEK293 cells: (**A**) cells transfected with TLR4-399C compared with TLR4-399C+TLR8-1A cells. IRF3 band intensity is higher in TLR4-399C+TLR8-1A double transfected cells under LPS, R8484, LPS+R848, Mtb RNA stimulation and unstimulated (US) cells than in cells transfected with TLR4-399C (**B**) in cells transfected with TLR4-399T compared with TLR4-399T+TLR8-1A cells, upon LPS or in unstimulated cells, band intensity is higher in double-transfected cells, whereas with R848 with or without LPS, it was slightly less (**C**) cells transfected with TLR4-399C+TLR8-1A compared with TLR4-399T+TLR8-1A cells. IRF3 band intensity is higher in TLR4-399C+TLR8-1A cells, irrespective of stimulation.

**Table 1 ijms-22-01560-t001:** Allele distributions of single nucleotide polymorphisms (SNPs) in the Indian cohort among controls and primary TB cases.

TLR SNPs (Nucleotide Change)	Alleles	N	Allele Frequency [N(%)] Controls Primary TB		OR [95% CI] *	*p*-Value
**TLR4-Asp299Gly (A > G)**	G	533	72 (27.48)	100 (33.22)	0.72 [0.49–1.07]	0.101
**TLR4-Thr399Ile (C > T)**	T	552	68 (23.37)	105 (31.44)	1.57 [1.04–2.36]	0.027
**TLR8-Met1Val (A > G)**	A	556	139 (47.60)	199 (58.70)	1.68 [1.08–2.63]	0.022
**TLR8-1, when TLR4-399CT/T**	A	395	34 (50.00)	59 (56.19)	1.97 [1.15–3.37]	0.013
**TLR8-1, when TLR4-399CC**	A	155	105 (47.09)	137 (59.83)	1.19 [0.52–2.72]	0.681

* Odds Ratios (ORs) based on Likelihood Ratio Tests (LRTs) adjusted for gender and age, as well as BCG status in case of TLR8.

**Table 2 ijms-22-01560-t002:** Allele distributions of SNPs in the Indian cohort among primary TB and relapse cases.

TLR SNPs	Alleles	N	Allele Frequency [N(%)] Primary TB Relapses		OR [95% CI] *	*p*-Value
**TLR4-Asp299Gly**	G	383	100 (33.22)	36 (39.13)	0.80 [0.49–1.32]	0.381
**TLR4-Thr399Ile**	T	376	105 (31.44)	33 (38.82)	1.36 [0.81–2.28]	0.242
**TLR8-Met1Val**	A	355	140 (58.70)	68 (72.34)	1.99 [1.03–3.82]	0.035
**TLR8-1, when TLR4-399CT/T**	A	111	59 (56.19)	25 (75.76)	2.90 [0.87–9.59]	0.069
**TLR8-1, when TLR4-399CC**	A	231	137 (59.82)	37 (71.15)	1.62 [0.69–3.81]	0.265

* ORs based on LRTs adjusted for gender and age, as well as BCG status in case of TLR8.

## Data Availability

Data available within the article or its Appendix A and on request from the authors.

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
