# Peer review of "Interaction of TLR4 and TLR8 in the Innate Immune Response against Mycobacterium Tuberculosis"

_ijms, 2021, doi:10.3390/ijms22041560_

Round 1

Reviewer 1 Report

Thada et al., International Journal of Molecular Sciences

General comments: 

The authors address the question of whether TLR4 and TLR8 function as a heterodimer.  The question is interesting and potentially important.  Although the data are suggestive of possible functional cross-talk between the two receptors, the data showing that TLR4 and TLR8 form a heterodimer or that this heterodimer has unique signaling capability are not strong.  Most of the data is obtained using HEK cells in which TLR4 and TLR8 constructs have been transiently “over-expressed”.  This model system is not well characterized making data interpretation difficult.  What are the expression levels of the individual TLRs?  How do they compare with immune cells?  How does “overexpression” impact the subcellular localization of the two receptors when expressed individually and together?  Does “overexpression” drive dimer formation in the absence of ligand?  What happens to adaptor proteins required for downstream signaling under these conditions?  It is not always clear whether the ancillary proteins, MD2 and CD14, which are required for LPS/TLR4 signaling are present.  Is it thought that LPS can signal/impact R848 signaling by working directly through TLR4, TLR8 or the hypothetical TLR4/TLR8 heterodimer in the absence of MD2/CD14?  In one experiment designed to demonstrate the presence of heterodimers, the authors claim that while mass spectrometric analysis of TLR8-HA immunoprecipitated from cell lysates identified the presence of TLR4 in 5 out of 6 experiments, the data “did not reach statistical significance”.   As cited below, there are several issues with data presentation and experimental approach.

Specific comments:

Fig. 3:  This experiment is not well described either in the text or in the legend.  What is the expression level of the various transfected proteins?  The authors suggest that 399T may not dimerize with TLR8, but the expression level of 399T appears to be considerably lower than that of 399C.  Is UNC93B1 required to demonstrate the co-ip between TL4 and TLR8?  These data need quantitation. There are no loading controls. The bands in panel C are not lined up with the other panels. What is the purpose of the data presented in the right-hand panel and what conclusion do the authors draw from this?  Are the authors drawing any conclusions from the data as to the role of UNC93 in promoting heterodimerization?  The unstimulated (US) TLR4 staining in panel b looks similar to the M.Tb lane.  Does this not suggest that the ability to co-ip does not require any stimulation?  US and NC and Input should be defined in the legend. 

Fig.4:  How are the authors interpreting these data?  We are only shown 1 or 2 cells at one time point.  The unstimulated cells show both TLR4 and TLR8 colocalized in vesicular structures.  Do these vesicles stain with endosomal markers?  Wouldn’t TLR4 be expected to localize to the plasma membrane?  What is the staining pattern at earlier times?  Do the HEK cells express CD14 or MD2?  Could this localization pattern be a result of overexpression of UNC, TLR4 or TLR8?  Where would TLR4 localize when expressed in the absence of TLR8 or MD2/CD14?  The figure legend mentions signaling but there are no signaling data presented.

Fig.5:  What is meant by co-localization frequency?  How is it measured?  The combination of LPS and R848 appears to be additive not synergistic as stated on p6 line199.

Fig, 6A:  Do these cells express both MD2 and CD14?  Are the TLR4 constructs localized to the cell surface?  The experiment needs a negative control. 

Fig. 6B:  Does “without accessory proteins” mean that there is no CD14 or MD2?  Therefore, these are not the same cells used in Fig. 6A?  Are the TLR4 constructs localized to the cell surface or to endosomes?  This should all be clarified in the Figure legend and the text.

Fig. 6C:  The figure legend is confusing.  There are no data to show that CLI-095 is inhibiting TLR4 signaling…although this is the presumption. The data indicate that NFkB activation in response to TLR8 ligands is enhanced in the presence of CLI-095. Do the authors interpret this to mean that TLR4 signaling is a negative regulator of TLR8 dependent activation of NFkB?

Fig. 6D:  This experiment does not support the idea that TLR4 signaling is regulating TLR8 dependent activation of NFkB. The data using R848 as a ligand are not interpretable as the control siRNA appears to be inhibiting NFkB activation.  In contrast, MTB activation of NFkB is not inhibited by either siRNA or siTLR4.

Fig. 6E and F:  The TLR4 expressing cells are not very responsive to LPS.  How are these data interpreted?

Fig. 7:  What is the rationale for looking at IRF3 protein levels as opposed to IRF3 activation?  How long are these cells treated with agonist?  Why would cells transfected with TLR4-399C/MD2 not express IRF3?  There are no loading controls on these blots.  

Author Response

General comment:

We want to thank this detailed referee for his detailed comments, questions and suggestions, as it really helped us to reshape and clarify some of the data that was previously missing. Examples for improved parts of the updated manuscript upon the suggestions of this referee are the more detailed report of Co-IP results (line 176-188) and discussion (392-398), about the role of MD2 and CD14 (line 404-411), quantification of Western Blots (Figure 7 and Figure 9) and the role of LPS-induced homodimerisation of TLR4 hindering heterodimerisation (line 411-413).

Point 1: Most of the data is obtained using HEK cells in which TLR4 and TLR8 constructs have been transiently “over-expressed”.  This model system is not well characterized making data interpretation difficult. How do they compare with immune cells? What are the expression levels of the individual TLRs?  How does “overexpression” impact the subcellular localization of the two receptors when expressed individually and together?  Does “overexpression” drive dimer formation in the absence of ligand?  What happens to adaptor proteins required for downstream signaling under these conditions?

Indeed the HEK-cell system is regarded as a ‘clean’ system, as HEK cells do not naturally express TLRs, but on the other hand, HEK cells are no immune cells, therefore they might react differently than native immune cells. Hence, we have also conducted experiments with THPs (Fig 7 C+D), as well as PBMC (Fig. 7 C+F). We included a part in the discussion, commenting about the artificiality of transfected HEK cells (404-407), and the possibility of spontaneous heterodimerisation (399-400, 418). Unfortunately, expression levels of TLRs have not been measured. However, we confirmed successful delivery to the endosome with confocal microscopy (Fig. 5)

Point 2: It is not always clear whether the ancillary proteins, MD2 and CD14, which are required for LPS/TLR4 signaling are present. 

We have to acknowledge that in the previous version of the manuscript, the issue of MD2 ane CD14 has been addressed insufficiently. We think, including it in the introduction (line 62-64), results (221, 231-232, 236), the figure legend (line 265-266) and discussion (404-4011) have indeed improved the manuscript.

Point 3: Is it thought that LPS can signal/impact R848 signaling by working directly through TLR4, TLR8 or the hypothetical TLR4/TLR8 heterodimer in the absence of MD2/CD14? 

In the absence of MD2/CD14, LPS responsiveness was low. However, with CD2/CD14, LPS reduced R848-responsiveness (line 184-188). This effect is now discussed in lines 411-413 and 427-429 and we want to thank the referee for pointing out that it was lacking in the previous version.

Point 4: In one experiment designed to demonstrate the presence of heterodimers, the authors claim that while mass spectrometric analysis of TLR8-HA immunoprecipitated from cell lysates identified the presence of TLR4 in 5 out of 6 experiments, the data “did not reach statistical significance”. 

We noted that indeed, we did not cite the p-value after Benjamin-Hochberg adjustment, and are therefore thankful that it was pointed out and corrected it accordingly (line 158-159).

Specific comments:

Figure 3:

Point 5: This experiment is not well described either in the text or in the legend. 

We thank the referee for bringing this up and changed the figure legend (line 199-209) and the description in the results part (176-181), as well as in the methods part (539-543).

Point 6: What is the expression level of the various transfected proteins?  These data need quantitation.

We appreciate the reviewer’s valid point and have hence quantified the data with Image J (Fig. 4 and Fig.9), which we included in the results (line 185-188 and 319-324) and discussion section (line 424-429-209 and 418-422 respectively).

Point 7: The authors suggest that 399T may not dimerize with TLR8, but the expression level of 399T appears to be considerably lower than that of 399C.

According to quantification data (Fig.4), which we conducted after the valuable input of the referee (see point 2), TLR4 399T expression levels were are actually higher than with 399C, but significantly reversed upon R848 stimulation (line 185-188). We commented on this in the revised version of the manuscript (line 424-429)

Point 8: Is UNC93B1 required to demonstrate the co-ip between TLR4 and TLR8?  Are the authors drawing any conclusions from the data as to the role of UNC93 in promoting heterodimerization? 

This is an important issue which, as we have to admit, was insufficiently addressed in the previous version of the manuscript. Although our data do not allow us to draw conclusions about an auxiliary function of Unc93 in the heterodimerisation of TLR4 and -8, we have included a discussion about this in the manuscript (line 394-398).

Point 9: There are no loading controls.

We thank the reviewer for the concern. We have missed to include the loading control in the previous version. Anti-GAPDH antibody was used as a loading control (~37 kDa) in native hTLR8HA+UNC93B1 native cells (updated in Fig.3f line 197) and in native HEK 293 cells (updated in fig.8h line 326)

Point 10: The bands in panel C are not lined up with the other panels.

We are sorry for the inconvenience; it is corrected in Fig.3 panel c (line 197).

Point 11: What is the purpose of the data presented in the right-hand panel and what conclusion do the authors draw from this?

Indeed, the figure itself and the legend might have been a little confusing in the previous version of the manuscript. We have altered it accordingly, changed the figure legend (line 199-209) and) and explained it in the results (line 176-179).

Point 12: The unstimulated (US) TLR4 staining in panel b looks similar to the M.Tb lane.  Does this not suggest that the ability to co-ip does not require any stimulation?

Response 8: We thank the referee for pointing out that our discussion in this regard has been insufficient.  Indeed there may be a potential for heterodimerisation without ligand binding. We extended the discussion section accordingly (line 392-94). We have also clarified the results itself (line 179-188).

Point 13: US and NC and Input should be defined in the legend. 

As requested by this referee, we have included the information in the revised version of the manuscript in line 201.

Fig.4 (Fig now modified to Fig.5):

Point 14: How are the authors interpreting these data?

In the previous version of the manuscript interpretation of the confocal scanning images may have been too brief. We are grateful that the referee indicated this and have included it accordingly in the discussion section (line 399-403). Furthermore, we improved the explanations in the results part (line 224-226).

Point 15: We are only shown 1 or 2 cells at one time point.  What is the staining pattern at earlier times?  

We have analysed colocalisation at 24 and 48h after transfection and 0.5, 1 and 2h after stimulation. Clearest results were obtained for 48h transfection and 2h stimulation, which was hence chosen for further analysis.

Point 16: The unstimulated cells show both TLR4 and TLR8 colocalized in vesicular structures.  Do these vesicles stain with endosomal markers?  Wouldn’t TLR4 be expected to localize to the plasma membrane?  Do the HEK cells express CD14 or MD2?  Could this localization pattern be a result of overexpression of UNC, TLR4 or TLR8?  Where would TLR4 localize when expressed in the absence of TLR8 or MD2/CD14? 

We acknowledge, that the topic of MD2 and CD14 was not addressed sufficiently in the previous version of the manuscript. Hence, we added it to the introduction (line 62-64), results (221, 231-232, 236), the figure legend (line 265-266) and discussion (404-4011) and believe that it contributed to the paper’s quality. Unfortunately, an endosomal marker was not used to stain the vesicles.

Point 17: The figure legend mentions signaling but there are no signaling data presented.

What was meant by ‘signaling’ was that, with Dynasore, downstream pathways of TLR4-TLR8 were blocked. We are sorry for the ambiguous use of the term and changed it accordingly (line 270-271).

Fig.5 (modified to Fig.6):

Point 18: What is meant by co-localization frequency?  How is it measured? 

Co-localisation is defined by spatial overlap of the fluorescent TLR4- and TLR8- labels, resulting a yellow signal in confocal microscopy. This explained in the new version of the manuscript (line 562-563).

Point 19: The combination of LPS and R848 appears to be additive not synergistic as stated on p6 line199.

We thank the reviewer for this very valid point and changed it accordingly (line 229)

Fig. 6A (modified to Fig.7A)

Point 20: Do these cells express both MD2 and CD14?  Are the TLR4 constructs localized to the cell surface? 

For LPS responsiveness, MD2 was transfected alongside TLR4, as indicated in the results (236), figure legend (290-291) and methods (542-543). The reviewer raises a valid point with regard to whether TLR4 is localised at the cell surface (added in the discussion (404-407)). In this experimental set-up, we think that LPS stimulates at the cell surface, but we cannot comment on endocytosis of TLR.

Point 21: The experiment needs a negative control. 

We are sorry for the inconvenience; the cells transfected with empty plasmid and stimulated with LPS are now included in figure 7A (line 281).

Fig. 6B (modified to Fig. 7B): 

Point 22: Does “without accessory proteins” mean that there is no CD14 or MD2?  Therefore, these are not the same cells used in Fig. 6A?

We have used the same cells used in Fig. 6A, but without MD2 and CD14 as per our modelling, CO-IP and mass spec results, the dimer formation can happen independently of accessory proteins in our experimental set-ups. We agree that this was not clearly stated and have included this in the discussion part of the revised version of the manuscript (404-406).

Point 23:  Are the TLR4 constructs localized to the cell surface or to endosomes?  This should all be clarified in the Figure legend and the text.

As mentioned above, we are thankful for bringing up the issue of explaining the use of MD2/CD14 and the localisation of TLR4 more thoroughly, as it was indeed insufficiently addressed in the previous version of the manuscript. Confocal studies have shown that with our experimental set-up, TLR4 localises both at the endosome and the surface, even without MD2/CD14. We have altered the figure legend accordingly (292-293), as well as extended the discussion on MD2/CD14 to clarify this point (404-406)

Fig. 6C (modified to Fig. 7C): 

Point 24: The figure legend is confusing.

Spurred by this referee, we amended the figure legend in order to enhance clarity (line 283-306).

Point 25: There are no data to show that CLI-095 is inhibiting TLR4 signaling…although this is the presumption.

LPS is a well-known ligand for TLR4 and there is good data that CLI-095 specifically blocks TLR4 (referenced now in the manuscript line 248-249). We were therefore satisfied with seeing LPS reactivity being reduced by CLI-095 and did not confirm again that indeed blockage of TLR4 by CLI-095 is the mechanisms of the reduction of LPS responsiveness.

Point 26: The data indicate that NFkB activation in response to TLR8 ligands is enhanced in the presence of CLI-095. Do the authors interpret this to mean that TLR4 signaling is a negative regulator of TLR8 dependent activation of NFkB?

We appreciate this comment and agree that the discussion needed clarification in this regard. As we saw spontaneous activation of IRF3 in HEK cells transfected with both TLR4 and -8, even without stimulation. Therefore, the interaction of TLR4 and -8 might lead a shift of the balance of the NFkB and Type I IFN-axes towards the latter, thereby reducing NFkB levels if both receptors are available, and increasing NFkB levels after TLR8 stimulation when no TLR4 is available comparatively. Accordingly, we clarified this hypothesis in the discussion section of the revised version of the manuscript (line 411-415)

Fig. 6D (modified to Fig. 7D): 

Point 27: This experiment does not support the idea that TLR4 signaling is regulating TLR8 dependent activation of NFkB. The data using R848 as a ligand are not interpretable as the control siRNA appears to be inhibiting NFkB activation.  In contrast, MTB activation of NFkB is not inhibited by either siRNA or siTLR4.

We agree that the siRNA-experiments do not strongly support our data, however, we decided to present them as they are to enable further discussion and interpretation to the wider community. However, the discussion has been improved, addressing potential confounders of this experiment (line 415-417), which might be due to the double transfection needed for siRNA, as well as, potentially, more effective inhibition of TLR4 by CLO-095 than siRNA.

Fig. 6E and F (modified to Fig. 7E and F): 

Point 28: The TLR4 expressing cells are not very responsive to LPS.  How are these data interpreted?

The PBMCs bearing TLR4 399C were less responsive than the PBMC bearing TLR4 399T, which is in line with previous findings of others, and is discussed in lines 373-377 of the revised version of the manuscript.

Fig. 7 (modified to Fig.8):

Point 29: What is the rationale for looking at IRF3 protein levels as opposed to IRF3 activation? 

Since it is a challenge to measure type I IFN directly we chose to look at the expression of IRF3 in line with the experiments described by Figure 3.

Point 30: How long are these cells treated with agonist? 

We agree with this referee that we were unclear in describing the experiment here. Cells were stimulated for 2h except for Mtb RNA, in which case stimulation took 16h, as specified in the legend of the revised version of the manuscript in line 331.

Point 31: Why would cells transfected with TLR4-399C/MD2 not express IRF3? 

This is a valid question, but we have to clarify that IRF3 experiments have been conducted without MD2, as other experimental set-ups have shown that TLR4 would localise to the endosome after transfection. As they are without MD2/CD14, LPS responsiveness by formation of a TLR4 homodimer is low in these cells (line 239-240).

Point 32: There are no loading controls on these blots.  

Indeed, we have missed to include the loading control in the previous version. Anti-GAPDH antibody was used as a loading control (~37 kDa) in native HEK 293 cells (updated in fig.8h line 326)

Reviewer 2 Report

Shruthi et al. reports that TLR4 plays a role in immune response against Mycobacterium tuberculosis (Mtb) and hypothesizes that endosomal interaction of TLR4 and TLR8 by forming a heterodimer regulates the tuberculosis (TB) immunity. In a SNP analysis of an Indian cohort, Shruthi et al. confirmed that both TLR4-399T and TLR8-1A conveying increased susceptibility towards TB, even though there is no established TLR4-ligand present in Mtb. Docking studies revealed that TLR4 and TLR8 can build a heterodimer, allowing interaction with TLR8 ligands. They also confirmed the interaction by mass spectrometry and co-immunoprecipitation and co-localization by confocal microscopy. Heterodimerization of TLR4 and TLR8 led to a strong IL12p40 signaling, as well as induction of NF-κB and IRF3. Thus, the interaction of TLR4 and TLR8 might open up new targets for vaccines or therapeutic drugs against Mtb. The reports are well-designed, organized, and descripted. But, several questions were occurred. I have minor questions.

  1. Please, result and discussion part should be described in detail, to increase reader’s understanding.
  2. There are minor typing errors in this paper, so please check the paper.
  3. I think it is better to clarify whether the expression of TLR SNPs is amino acid or nucleotide in the text. It is confusing whether T of TLR4-399T is Thr or Ile.
  4. Please, convert the western blot data to graph format by image J to increase reader’s understanding.
  5. Compared to Figure 3 and Figure 5, the results of Co-IP and Co-localization seem to be different in the treatment of LPS+R848, why?
  6. You explained TLR4 and TLR8 interaction is important in TB. I wonder interaction between TLR4 and TLR8 is important for eliminating the TB.    
  7. MDR and XDR-TB is serious problem in TB therapy. I wonder interaction of TLR4 and TLR8 occurs these 2 cases of TB.

[2.4. Co-Immunoprecipitation]

  1. When TLR4 and TLR8 form a heterodimer, why does the treatment of the TLR4 agonist LPS not respond as strongly as the treatment of the TLR8 agonist, R848?
  2. LPS stimulated cells showed TLR4 band, so why does LPS+R848 stimulated cells show TLR4 band weaker than R848 stimulated cells?
  3. Please rearrange the data of Figure 3. It is confused which antibody is used for immunoprecipitation and which protein is detected in immunoblotting.

[2.5. Co-localisation]

  1. I think it is better to add the reason why ssRNA40 is used as stimulant in Figure 4.

[2.6. Functional studies]

  1. Figure 6C shows an increase in NF-κB response by TLR8 ligand when TLR4 is inhibited, but why does not NF-κB response increase when TLR4 is silenced by siRNA in Figure 6D?
  2. In Figure 7, It seems that there is no difference between stimulated and unstimulated cells. Then, does the interaction of TLR4 and TLR8 produce type I IFN without the presence of TLR4 and TLR8 ligands?

Author Response

Please, result and discussion part should be described in detail, to increase reader’s understanding.

We agree, that in the first version results and discussion were written very briefly and concise, which may lead to problems in understanding. We thoroughly rewrote both sections and think that it was improved by this. Examples for a more detailed description are and improve report of Co-IP results (line 176-188) and a respective discussion (392-398), the quantification of Western Blots (Figure 7 and Figure 9), a discussion about the role of accessory proteins of TLR4 (line 404-411),  and the role of LPS-induced homodimerisation of TLR4 hindering heterodimerisation (line 411-413).

Point 1: There are minor typing errors in this paper, so please check the paper.

We checked the paper for typos and corrected all typing errors as indicated by the highlighted lines (i.e. line 116, 168, 360, 372, 387, 389).

Point 2: I think it is better to clarify whether the expression of TLR SNPs is amino acid or nucleotide in the text. It is confusing whether T of TLR4-399T is Thr or Ile.

We agree that in the first version nomenclature was written confusingly and we enhanced clarity by only using the three-letter abbreviations for the amino acids and have clearly indicated now that the abbreviations C/T/A/G stand for the nucleic acids (line 100 and 101). We also indicated the nucleotide change in addition to the amino acid change in Table 1 (line 123), and added the nucleotide nomenclature of the SNP to the figure legend of figure 1 (line 143).

Point 3: Please convert the Western Blot data to graph format by image J to increase reader’s understanding.

We thank this reviewer for the suggestion and have quantified the Western Blot result using image J. This is shown now in Fig. 4 and 9 (line 210-218 and 333-342, respectively), and is also described in the results section line 185-188 and 319-324.

Point 4: Compared to Figure 3 and Figure 5, the results of Co-IP and Co-localization seem to be different in the treatment of LPS + R848, why?

We are grateful that this referee pointed this out. The reason for the difference may be that the read-out of the two methods is different, and due to the proximity of the two receptors, there might be co-localization without actual heterodimerisation, leading to the different results. We added this subject within the discussion section (line 399-403).

Point 5: You explained TLR4 and TLR8 interaction is important in TB. I wonder interaction between TLR4 and TLR8 is important for eliminating the TB.

We appreciated this interesting point of discussion and accordingly added a paragraph to our discussion section (line 445-450)

Point 6: MDR and XDR-TB is serious problem in TB therapy. I wonder interaction of TLR4 and TLR8 occurs these 2 cases of TB.

MDR and XDR is a serious problem, indeed, and in addition to point 5, we added this important aspect to the discussion (line 450). We don´t think that our findings will solve the MDR-problem, however, we thank the referee for this valuable impulse.

Point 7: When TLR4 and TLR8 form a heterodimer, why does the treatment of the TLR4 agonist LPS not respond as strongly as the treatment of the TLR8 agonist, R848?

We thank this reviewer for this question, which is important and may add to the understanding and interpretation of our data. We have therefore included a discussion of this result in the manuscript (line 411-413).

Point 8: LPS stimulated cells showed TLR4 band, so why does LPS+R848 stimulated cells show TLR4 band weaker than R848 stimulated cells?

Pointing out this difference in experimental results helped us shaping the respective sections in the results (line 184-188) and discussion (411-413 and 427-429) part. Therefore, we are glad the referee brought this up, as it enhanced the clarity of the manuscript.

Point 9: Please rearrange the data of Figure 3. It is confused which antibody is used for immunoprecipitation and which protein is detected in immunoblotting.

We agree that in the old version the data may have been confusing. We rearranged Figure 3 as suggested (line 197-209).

Point 10: I think it is better to add the reason why ssRNA40 is used as stimulant in Figure 4.

This is a valid point and the reason for using ssRNA40 was added to the revised version of the manuscript (line 222-224).

Point 11. Figure 6C shows an increase in NF-κB response by TLR8 ligand when TLR4 is inhibited, but why does not NF-κB response increase when TLR4 is silenced by siRNA in Figure 6D?

We agree that this potential discrepancy has not been discussed/explained well in the old version of the manuscript. We discussed this in-depth in the discussion section of the revised version of the manuscript (lines 415-417). In brief, when comparing R848 and MTB-RNA stimulation, after treatment with siTLR4 and control-si, no difference was observed (Fig. 7D). Although, this was the case with CLI-95 treated cells (Fig 7C). This might be due to technical difficulties well known to experiments with siRNA and that CLI-95 might give a more complete blockage of TLR4 signaling than siRNA.

Point 12: In Figure 7, It seems that there is no difference between stimulated and unstimulated cells. Then, does the interaction of TLR4 and TLR8 produce type I IFN without the presence of TLR4 and TLR8 ligands?

We acknowledge that in the old version of the manuscript, the data were confusing. The results part (line 317-324) and discussion (line 418-422) were changed in order to clarify this point. In brief, with WB quantification (Fig.9) there was little difference in stimulated and unstimulated in cells transfected with TLR4-399C/TLR8- 1A. This might be hinting at spontaneous heterodimerisation, supported by the data, that expression levels are higher with TLR4-399C than 399T.

Reviewer 3 Report

The manuscript titled "Interaction of TLR4 and TLR8 in the innate immune response against Mycobacterium tuberculosis" is well written with interesting findings.

There are no specific comments for the authors.

Author Response

We thank this referee for the positive feedback.